# How To Make the Gradients Small Stochastically: Even Faster Convex and Nonconvex SGD*

**Zeyuan Allen-Zhu**
Microsoft Research AI
Redmond, WA 98052
zeyuan@csail.mit.edu

## Abstract

Stochastic gradient descent (SGD) gives an optimal convergence rate when minimizing convex stochastic objectives $f(x)$. However, in terms of making the gradients small, the original SGD does not give an optimal rate, even when $f(x)$ is convex.

If $f(x)$ is convex, to find a point with gradient norm $\varepsilon$, we design an algorithm SGD3 with a near-optimal rate $\widetilde{O}(\varepsilon^{-2})$, improving the best known rate $O(\varepsilon^{-8/3})$ of [17]. If $f(x)$ is nonconvex, to find its $\varepsilon$-approximate local minimum, we design an algorithm SGD5 with rate $\widetilde{O}(\varepsilon^{-3.5})$, where previously SGD variants only achieve $\widetilde{O}(\varepsilon^{-4})$ [6, 14, 30]. This is no slower than the best known stochastic version of Newton's method in all parameter regimes [27].

## 1 Introduction

In convex optimization and machine learning, the classical goal is to design algorithms to decrease objective values, that is, to find points $x$ with $f(x) - f(x^*) \leq \varepsilon$. In contrast, the *rate of convergence for the gradients*, that is,

the number of iterations $T$ needed to find a point $x$ with $\|\nabla f(x)\| \leq \varepsilon$,

is a harder problem and sometimes needs new algorithmic ideas [25]. For instance, in the full-gradient setting, accelerated gradient descent alone is suboptimal for this new goal, and one needs additional tricks to get the fastest rate [25]. We review these tricks in Section 1.1.

In the convex (online) *stochastic* optimization, to the best of our knowledge, tight bounds are not yet known for finding points with small gradients. The best recorded rate was $T \propto \varepsilon^{-8/3}$ [17], and it was raised as an open question [1] regarding how to improve it.

In this paper, we design two new algorithms, SGD2 which gives rate $T \propto \varepsilon^{-5/2}$ using Nesterov's tricks, and SGD3 which gives an even better rate $T \propto \varepsilon^{-2} \log^3 \frac{1}{\varepsilon}$ which is optimal up to log factors.

**Motivation.** Studying the rate of convergence for the minimizing gradients can be important at least for the following two reasons.

- In many situations, points with small gradients fit better our final goals.

Nesterov [25] considers the dual approach for solving constrained minimization problems. He argued that "the gradient value $\|\nabla f(x)\|$ serves as the measure of feasibility and optimality of the primal solution," and thus is the better goal for minimization purpose.[2]

In matrix scaling [7, 10], given a non-negative matrix, one wants to re-scale its rows and columns to make it doubly stochastic. This problem has been applied in image reconstruction, operations research, decision and control, and other scientific disciplines (see survey [20]). The goal for matrix scaling is to find points with small gradients, but not small objectives.

- Designing algorithms to find points with small gradients can help us understand non-convex optimization better and design faster non-convex machine learning algorithms.

  Without strong assumptions, non-convex optimization theory is always in terms of finding points with small gradients (i.e., approximate stationary points or local minima). Therefore, to understand non-convex stochastic optimization better, perhaps we should first figure out the best rate for *convex* stochastic optimization. In addition, if new algorithmic ideas are needed, can we also apply them to the non-convex world? We find positive answers to this question, and also obtain better rates for standard non-convex optimization tasks.

## 1.1 Review: Prior Work on Deterministic Convex Optimization

Suppose $f(x)$ is a Lipschitz smooth convex function with smoothness parameter $L$. Then, it is well-known that accelerated gradient descent (AGD) [23, 24] finds a point $x$ satisfying $f(x) - f(x^*) \leq \delta$ using $T = O(\frac{\sqrt{L}}{\sqrt{\delta}})$ gradient computations of $\nabla f(x)$. To turn this into a gradient guarantee, we can apply the smoothness property of $f(x)$ which gives $\|\nabla f(x)\|^2 \leq L(f(x) - f(x^*))$. This means

$$\text{AGD converges in rate } T \propto \frac{L}{\varepsilon}.$$

Nesterov [25] proposed two different tricks to improve upon such rate.

**Nesterov's First Trick: GD After AGD.** Recall that starting from a point $x_0$, if we perform $T$ steps of gradient descent (GD) $x_{t+1} = x_t - \frac{1}{L}\nabla f(x_t)$, then it satisfies $\sum_{t=0}^{T-1} \|\nabla f(x_t)\|^2 \leq L(f(x_0) - f(x^*))$. In addition, if this $x_0$ is already the output of AGD for another $T$ iterations, then it satisfies $f(x_0) - f(x^*) \leq O(\frac{L}{T^2})$. Putting the two inequalities together, we have $\min_{t=0}^{T-1}\left\{\|\nabla f(x_t)\|^2\right\} \leq O(\frac{L^2}{T^3})$. We call this method "GD after AGD," and

$$\text{"GD after AGD" converges in rate } T \propto \frac{L^{2/3}}{\varepsilon^{2/3}}.$$

**Nesterov's Second Trick: AGD After Regularization.** Alternatively, we can also regularize $f(x)$ by defining $g(x) = f(x) + \frac{\sigma}{2}\|x - x_0\|^2$. This new function $g(x)$ is $\sigma$-strongly convex, so AGD converges *linearly*, meaning that using $T \propto \frac{\sqrt{L}}{\sqrt{\sigma}} \log \frac{L}{\varepsilon}$ gradients we can find a point $x$ satisfying $\|\nabla g(x)\|^2 \leq L(g(x) - g(x^*)) \leq \varepsilon^2$. If we choose $\sigma \propto \varepsilon$, then this implies $\|\nabla f(x)\| \leq \|\nabla g(x)\| + \varepsilon \leq 2\varepsilon$. We call this method "AGD after regularization," and

$$\text{"AGD after regularization" converges in rate } T \propto \frac{L^{1/2}}{\varepsilon^{1/2}} \log \frac{L}{\varepsilon}.$$

This is optimal up to a log factor, because first-order methods need $T = \Omega(\sqrt{L/\delta})$ gradient computations to find $f(x) - f(x^*) \leq \delta$ [23], but $f(x) - f(x^*) \leq \|\nabla f(x)\| \cdot \|x - x^*\| \leq O(\|\nabla f(x)\|)$.

## 1.2 Our Results: Stochastic Convex Optimization

Consider the stochastic setting where the convex objective $f(x) := \mathbb{E}_i[f_i(x)]$ and the algorithm can only compute stochastic gradients $\nabla f_i(x)$ at any point $x$ for a random $i$. Let $T$ be the number of stochastic gradient computations. It is well-known that stochastic gradient descent (SGD) finds a point $x$ with $f(x) - f(x^*) \leq \delta$ in (see for instance textbooks [8, 18, 26])

$$T = O\left(\frac{\mathcal{V}}{\delta^2}\right) \text{ iterations} \quad \text{or} \quad T = O\left(\frac{\mathcal{V}}{\sigma\delta}\right) \text{ if } f(x) \text{ is } \sigma\text{-strongly convex.}$$

| | **algorithm** | | **gradient complexity** $T$ | | **2nd-order smooth** |
|---|---|---|---|---|---|
| online convex | `SGD` | (naive) | $O(\varepsilon^{-4})$ | (folklore, see Theorem 4.2) | |
| | `SGD1` | (SGD after SGD) | $O(\varepsilon^{-8/3})$ | (see [17] or Theorem 1) | |
| | `SGD2` | (SGD after regularization) | $O(\varepsilon^{-5/2})$ | (see Theorem 2) | |
| | `SGD3` | (SGD + recursive regularization) | $O(\varepsilon^{-2} \cdot \log^3 \frac{1}{\varepsilon})$ | (see Theorem 3) | |
| online strongly convex | `SGD`$^{\text{sc}}$ | (naive) | $O(\varepsilon^{-2} \cdot \kappa)$ | (see Theorem 4.2) | no |
| | `SGD1`$^{\text{sc}}$ | (SGD after SGD) | $O(\varepsilon^{-2} \cdot \kappa^{1/2})$ | (see Theorem 1) | |
| | `SGD3`$^{\text{sc}}$ | (SGD + recursive regularization) | $O(\varepsilon^{-2} \cdot \log^3 \kappa)$ | (see Theorem 3) | |
| online nonconvex ($\sigma$-nonconvex) | `SGD` | (naive) | $O(\varepsilon^{-4})$ | (see [16]) | |
| | `SCSG` | | $O(\varepsilon^{-10/3})$ | (see [21]) | |
| | `SGD4` | | $O(\varepsilon^{-2} + \sigma\varepsilon^{-4})$ | (see Theorem 4) | |
| | `Natasha1.5` | | $O(\varepsilon^{-3} + \sigma^{1/3}\varepsilon^{-10/3})$ | (see [3]) | |
| | `SGD variants` | | $\widetilde{O}(\varepsilon^{-4})$ | (see [6, 14, 30]) | |
| | `SGD5` | | $\widetilde{O}(\varepsilon^{-3.5})$ | (see Theorem 5) | needed |
| | cubic Newton | | $\widetilde{O}(\varepsilon^{-3.5})$ | (see [27]) | |
| | `Natasha2` | | $O(\varepsilon^{-3.25})$ | (see [3]) | |

Table 1: Comparison of first-order *online stochastic* methods for finding $\|\nabla f(x)\| \le \varepsilon$. Following tradition, in these bounds, we hide variance and smoothness parameters in big-$O$ and only show the dependency on $\varepsilon$, the condition number $\kappa = \frac{L}{\sigma} \ge 1$ (if the objective is $\sigma$-strongly convex), or the nonconvexity parameter $\sigma$.

Both rates are asymptotically optimal in terms of decreasing objective, and $\mathcal{V}$ is an absolute bound on the variance of the stochastic gradients. Using the same argument $\|\nabla f(x)\|^2 \le L(f(x) - f(x^*))$ as before, SGD finds a point $x$ with $\|\nabla f(x)\| \le \varepsilon$ in

$$T = O\big(\tfrac{L^2\mathcal{V}}{\varepsilon^4}\big) \text{ iterations} \quad \text{or} \quad T = O\big(\tfrac{L\mathcal{V}}{\sigma\varepsilon^2}\big) \text{ if } f(x) \text{ is } \sigma\text{-strongly convex.} \qquad \text{(SGD)}$$

These rates are not optimal. We investigate three approaches to improve such rates.

**New Approach 1: SGD after SGD.** Recall in Nesterov's first trick, he replaced the use of the inequality $\|\nabla f(x)\|^2 \le L(f(x) - f(x^*))$ by $T$ steps of gradient descent. In the stochastic setting, can we replace this inequality with $T$ steps of SGD? We call this algorithm SGD1 and prove that

**Theorem 1** (informal). *For convex stochastic optimization, SGD1 finds $x$ with $\|\nabla f(x)\| \le \varepsilon$ in*

$$T = O\Big(\frac{L^{2/3}\mathcal{V}}{\varepsilon^{8/3}}\Big) \text{ iterations} \quad \text{or} \quad T = O\Big(\frac{L^{1/2}\mathcal{V}}{\sigma^{1/2}\varepsilon^2}\Big) \text{ if } f(x) \text{ is } \sigma\text{-strongly convex.} \qquad \text{(SGD1)}$$

The rate $T \propto \varepsilon^{-8/3}$, in the special case of unconstrained minimization, was first discovered by Ghadimi and Lan [17] using a more complicated algorithm. The rate $T \propto \frac{1}{\sigma^{1/2}\varepsilon^2}$ does not seem to be known before.

**New Approach 2: SGD after regularization.** Recall that in Nesterov's second trick, he defined $g(x) = f(x) + \frac{\sigma}{2}\|x - x_0\|^2$ as a regularized version of $f(x)$, and applied the strongly-convex version of AGD to minimize $g(x)$. Can we apply this trick to the stochastic setting?

Note the parameter $\sigma$ has to be on the magnitude of $\varepsilon$ because $\nabla g(x) = \nabla f(x) + \sigma(x - x_0)$ and we wish to make sure $\|\nabla f(x)\| = \|\nabla g(x)\| \pm \varepsilon$. Therefore, if we apply SGD1 to minimize $g(x)$ to find a point $\|\nabla g(x)\| \le \varepsilon$, the convergence rate is $T \propto \frac{1}{\sigma^{1/2}\varepsilon^2} = \frac{1}{\varepsilon^{2.5}}$. We call this algorithm SGD2.

**Theorem 2** (informal). *For convex stochastic optimization, SGD2 finds $x$ with $\|\nabla f(x)\| \le \varepsilon$ in*

$$T = O\big(\tfrac{L^{1/2}\mathcal{V}}{\varepsilon^{5/2}}\big) \text{ iterations.} \qquad \text{(SGD2)}$$

Again, this $T \propto \frac{1}{\varepsilon^{5/2}}$ rate does not seem to be known before.

**New Approach 3: SGD and recursive regularization.** In the second approach above, the $\varepsilon^{0.5}$ sub-optimality gap is due to the choice of $\sigma \propto \varepsilon$ which ensures $\|\sigma(x - x_0)\| \le \varepsilon$.

Intuitively, if $x_0$ were sufficiently close to $x^*$ (and thus were also close to the approximate minimizer $x$), then we could choose $\sigma \gg \varepsilon$ so that $\|\sigma(x - x_0)\| \leq \varepsilon$ still holds. In other words, an appropriate *warm start* $x_0$ could help us break the $\varepsilon^{-2.5}$ barrier and get a better convergence rate. However, how to find such $x_0$? We find it by constructing a "less warm" starting point and so on. This process is summarized by the following algorithm which recursively finds the warm starts.

Starting from $f^{(0)}(x) := f(x)$, we define $f^{(s)}(x) := f^{(s-1)}(x) + \frac{\sigma_s}{2}\|x - \widehat{x}_s\|^2$ where $\sigma_s = 2\sigma_{s-1}$ and $\widehat{x}_s$ is an approximate minimizer of $f^{(s-1)}(x)$ that is simply calculated from the naive SGD. We call this method SGD3, and prove that

**Theorem 3** (informal). *For convex stochastic optimization, SGD3 finds $x$ with $\|\nabla f(x)\| \leq \varepsilon$ in*

$$T = O\big(\tfrac{\log^3(L/\varepsilon)\cdot\mathcal{V}}{\varepsilon^2}\big) \text{ iterations} \quad or \quad T = O\big(\tfrac{\log^3(L/\sigma)\cdot\mathcal{V}}{\varepsilon^2}\big) \text{ if } f(x) \text{ is } \sigma\text{-strongly convex. (SGD3)}$$

Our new rates in Theorem 3 not only improve the best known result of $T \propto \varepsilon^{-8/3}$, but also are near optimal because $\Omega(\mathcal{V}/\varepsilon^2)$ is clearly a lower bound: even to decide whether a point $x$ has $\|\nabla f(x)\| \leq \varepsilon$ or $\|\nabla f(x)\| > 2\varepsilon$ requires $\Omega(\mathcal{V}/\varepsilon^2)$ samples of the stochastic gradient. Perhaps interestingly, our dependence on the smoothness parameter $L$ (or the condition number $\kappa := L/\sigma$ if strongly convex) is only polylogarithmic, as opposed to polynomial in all previous results.

## 1.3 Roadmap

We introduce notions in Section 2 and formalize the convex problem in Section 3. We review classical (convex) SGD theorems with objective decrease in Section 4. We give an auxiliary lemma in Section 5 show our SGD3 results in Section 6. We apply our techniques to non-convex optimization and give algorithms SGD4 and SGD5 in Section 7. We discuss more related work in Appendix A, and show our results on SGD1 and SGD2 respectively in Appendix B and Appendix C.

## 2 Preliminaries

Throughout this paper, we denote by $\|\cdot\|$ the Euclidean norm. We use $i \in_R [n]$ to denote that $i$ is generated from $[n] = \{1, 2, \ldots, n\}$ uniformly at random. We denote by $\nabla f(x)$ the gradient of function $f$ if it is differentiable, and $\partial f(x)$ any subgradient if $f$ is only Lipschitz continuous. We denote by $\mathbb{I}[event]$ the indicator function of probabilistic events.

We denote by $\|\mathbf{A}\|_2$ the spectral norm of matrix $\mathbf{A}$. For symmetric matrices $\mathbf{A}$ and $\mathbf{B}$, we write $\mathbf{A} \succeq \mathbf{B}$ to indicate that $\mathbf{A} - \mathbf{B}$ is positive semidefinite (PSD). Therefore, $\mathbf{A} \succeq -\sigma\mathbf{I}$ if and only if all eigenvalues of $\mathbf{A}$ are no less than $-\sigma$. We denote by $\lambda_{\min}(\mathbf{A})$ and $\lambda_{\max}(\mathbf{A})$ the minimum and maximum eigenvalue of a symmetric matrix $\mathbf{A}$.

Recall some definitions on strong convexity and smoothness (and they have other equivalent definitions, see textbook [23]).

**Definition 2.1.** *For a function $f\colon \mathbb{R}^d \to \mathbb{R}$,*

- *$f$ is $\sigma$-**strongly convex** if $\forall x, y \in \mathbb{R}^d$, it satisfies $f(y) \geq f(x) + \langle\partial f(x), y - x\rangle + \frac{\sigma}{2}\|x - y\|^2$.*
- *$f$ is of $\sigma$-**bounded nonconvexity** (or $\sigma$-**nonconvex** for short) if $\forall x, y \in \mathbb{R}^d$, it satisfies $f(y) \geq f(x) + \langle\partial f(x), y - x\rangle - \frac{\sigma}{2}\|x - y\|^2$. [3]*
- *$f$ is $L$-**Lipschitz smooth** (or $L$-**smooth** for short) if $\forall x, y \in \mathbb{R}^d$, $\|\nabla f(x) - \nabla f(y)\| \leq L\|x - y\|$.*
- *$f$ is $L_2$-**second-order smooth** if $\forall x, y \in \mathbb{R}^d$, it satisfies $\|\nabla^2 f(x) - \nabla^2 f(y)\|_2 \leq L_2\|x - y\|$.*

**Definition 2.2.** *For composite function $F(x) = \psi(x) + f(x)$ where $\psi(x)$ is proper convex, given a parameter $\eta > 0$, the **gradient mapping** of $F(\cdot)$ at point $x$ is*

$$\mathcal{G}_{F,\eta}(x) := \frac{1}{\eta}\big(x - x^+\big) \quad where \quad x^+ = \arg\min_y \big\{\psi(y) + \langle\nabla f(x), y\rangle + \frac{1}{2\eta}\|y - x\|^2\big\}$$

*In particular, if $\psi(\cdot) \equiv 0$, then $\mathcal{G}_{F,\eta}(x) \equiv \nabla f(x)$.*

Recall the following property about gradient mapping —see for instance [29, Lemma 3.7])

**Lemma 2.3.** *Let* $F(x) = \psi(x) + f(x)$ *where* $\psi(x)$ *is proper convex and* $f(x)$ *is* $\sigma$-*strongly convex and* $L$-*smooth. For every* $x, y \in \{x \in \mathbb{R}^d : \psi(x) < +\infty\}$, *letting* $x^+ = x - \eta \cdot \mathcal{G}_{F,\eta}(x)$, *we have*

$$\forall \eta \in \left(0, \frac{1}{L}\right]: \quad F(y) \geq F(x^+) + \langle \mathcal{G}_{F,\eta}(x), y - x \rangle + \frac{\eta}{2} \|\mathcal{G}_{F,\eta}(x)\|^2 + \frac{\sigma}{2}\|y - x\|^2 \ .$$

The following definition and properties of Fenchel dual for convex functions is classical, and can be found for instance in the textbook [26].

**Definition 2.4.** *Given proper convex function* $h(y)$, *its Fenchel dual* $h^*(\beta) := \max_y\{y^\top \beta - h(y)\}$.

**Proposition 2.5.** $\nabla h^*(\beta) = \arg\max_y\{y^\top \beta - h(y)\}$.

**Proposition 2.6.** *If* $h(\cdot)$ *is* $\sigma$-*strongly convex, then* $h^*(\cdot)$ *is* $\frac{1}{\sigma}$-*smooth.*

# 3  Problem Formalization

Throughout this paper (except our nonconvex application Section 7), we minimize *convex* stochastic composite objective:

$$\min_{x \in \mathbb{R}^d} \left\{ F(x) = \psi(x) + f(x) := \psi(x) + \tfrac{1}{n} \sum_{i \in [n]} f_i(x) \right\} \ , \tag{3.1}$$

where

1. $\psi(x)$ is proper convex (a.k.a. the proximal term),
2. $f_i(x)$ is differentiable for every $i \in [n]$,
3. $f(x)$ is $L$-smooth and $\sigma$-strongly convex for some $\sigma \in [0, L]$ that could be zero,
4. $n$ can be very large of even infinite (so $f(x) = \mathbb{E}_i[f_i(x)]$),[4] and
5. the stochastic gradients $\nabla f_i(x)$ have a bounded variance (over the domain of $\psi(\cdot)$), that is

$$\forall x \in \{y \in \mathbb{R}^d \,|\, \psi(y) < +\infty\}: \quad \mathbb{E}_{i \in_R [n]} \|\nabla f(x) - \nabla f_i(x)\|^2 \leq \mathcal{V} \ .$$

We emphasize that the above assumptions are all classical.

In the rest of the paper, we define $T$, the gradient complexity, as the number of computations of $\nabla f_i(x)$. We search for points $x$ so that the gradient mapping $\|\mathcal{G}_{F,\eta}(x)\| \leq \varepsilon$ for any $\eta \approx \frac{1}{L}$. Recall from Definition 2.2 that if there is no proximal term (i.e., $\psi(x) \equiv 0$), then $\mathcal{G}_{F,\eta}(x) = \nabla f(x)$ for any $\eta > 0$. We want to study the best tradeoff between the gradient complexity $T$ and the error $\varepsilon$.

We say an algorithm is *online* if its gradient complexity $T$ is independent of $n$. This tackles the big-data scenario when $n$ is extremely large or even infinite (i.e., $f(x) = \mathbb{E}_i[f_i(x)]$ for some random variable $i$). The stochastic gradient descent (SGD) method and all of its variants studied in this paper are online. In contrast, GD, AGD [23, 24], and Katyusha [2] are offline methods because their gradient complexity depends on $n$ (see Table 2 in appendix).

# 4  Review: SGD with Objective Value Convergence

Recall that stochastic gradient descent (SGD) repeatedly performs *proximal updates* of the form

$$x_{t+1} = \arg\min_{y \in \mathbb{R}^d}\{\psi(y) + \tfrac{1}{2\alpha}\|y - x_t\|^2 + \langle \nabla f_i(x_t), y\rangle\} \ ,$$

where $\alpha > 0$ is some learning rate, and $i$ is chosen in $1, 2, \ldots, n$ uniformly at random per iteration. Note that if $\psi(y) \equiv 0$ then $x_{t+1} = x_t - \alpha \nabla f_i(x_t)$. For completeness' sake, we summarize it in Algorithm 1. If $f(x)$ is also known to be strongly convex, to get the tightest convergence rate, one can repeatedly apply SGD with decreasing learning rate $\alpha$ [19]. We summarize this algorithm as SGD$^{\text{sc}}$ in Algorithm 2.

The following theorem describes the rates of convergence in objective values for SGD and SGD$^{\text{sc}}$ respectively. Their proofs are classical (and included in Appendix D); however, for our exact statements, we cannot find them recorded anywhere.[5]

**Algorithm 1** $\mathtt{SGD}(F, x_0, \alpha, T)$

---

**Input:** function $F(x) = \psi(x) + \frac{1}{n}\sum_{i=1}^{n} f_i(x)$; initial vector $x_0$; learning rate $\alpha > 0$; $T \geq 1$.
  $\diamond$ *if $f(x) = \frac{1}{n}\sum_{i=1}^{n} f_i(x)$ is L-smooth, optimal choice $\alpha = \Theta\big(\min\big\{\frac{\|x_0 - x^*\|}{\sqrt{\mathcal{V}T}}, \frac{1}{L}\big\}\big)$*
1: **for** $t = 0$ **to** $T - 1$ **do**
2:    $i \leftarrow$ a random index in $[n]$;
3:    $x_{t+1} \leftarrow \arg\min_{y \in \mathbb{R}^d}\{\psi(y) + \frac{1}{2\alpha}\|y - x_t\|^2 + \langle\nabla f_i(x_t), y\rangle\}$;
4: **return** $\overline{x} = \frac{x_1 + \cdots + x_T}{T}$.

---

---
**Algorithm 2** $\mathtt{SGD^{sc}}(F, x_0, \sigma, L, T)$

---

**Input:** function $F(x) = \psi(x) + \frac{1}{n}\sum_{i=1}^{n} f_i(x)$; initial vector $x_0$; parameters $0 < \sigma \leq L$; $T \geq \frac{L}{\sigma}$.
  $\diamond$ *$f(x)$ is $\sigma$-strongly convex and $f(x) = \frac{1}{n}\sum_{i=1}^{n} f_i(x)$ is L-smooth*
1: **for** $t = 1$ **to** $N = \lfloor\frac{T}{8L/\sigma}\rfloor$ **do**    $x_t \leftarrow \mathtt{SGD}\big(F, x_{t-1}, \frac{1}{2L}, \frac{4L}{\sigma}\big)$;
2: **for** $k = 1$ **to** $K = \lfloor\log_2(\sigma T/16L)\rfloor$ **do**    $x_{N+k} \leftarrow \mathtt{SGD}\big(F, x_{N+k-1}, \frac{1}{2^k L}, \frac{2^{k+2}L}{\sigma}\big)$;
3: **return** $\overline{x} = x_{N+K}$.

---

**Theorem 4.1.** *Let $x^* \in \arg\min_x\{F(x)\}$. To solve Problem (3.1) given a starting vector $x_0 \in \mathbb{R}^d$,*

*(a)* $\mathtt{SGD}(F, x_0, \alpha, T)$ *outputs $\overline{x}$ satisfying $\mathbb{E}[F(\overline{x})] - F(x^*) \leq \frac{\alpha\mathcal{V}}{2(1-\alpha L)} + \frac{\|x_0 - x^*\|^2}{2\alpha T}$ as long as $\alpha < 1/L$. In particular, if $\alpha$ is tuned optimally, it satisfies*

$$\mathbb{E}[F(\overline{x})] - F(x^*) \leq O\big(\frac{L\|x_0 - x^*\|^2}{T} + \frac{\sqrt{\mathcal{V}}\|x_0 - x^*\|}{\sqrt{T}}\big) \ .$$

*(b) If $f(x)$ is $\sigma$-strongly convex and $T \geq \frac{L}{\sigma}$, then $\mathtt{SGD^{sc}}(F, x_0, \sigma, L, T)$ outputs $\overline{x}$ satisfying*

$$\mathbb{E}[F(\overline{x})] - F(x^*) \leq O\big(\frac{\mathcal{V}}{\sigma T}\big) + \big(1 - \frac{\sigma}{L}\big)^{\Omega(T)}\sigma\|x_0 - x^*\|^2 \ .$$

As a sanity check, if $\mathcal{V} = 0$, the convergence rate of SGD matches that of GD. (However, if $\mathcal{V} = 0$, one can apply accelerated gradient descent of Nesterov [22, 23] instead for a faster rate.)

To turn Theorem 4.1 into a rate of convergence for the gradients, we can simply apply Lemma 2.3 which implies

$$\forall\eta \in \big(0, \frac{1}{L}\big]: \quad \frac{\eta}{2}\|\mathcal{G}_{F,\eta}(\overline{x})\|^2 \leq F(\overline{x}) - F(\overline{x}^+) \leq F(\overline{x}) - F(x^*) \ . \tag{4.1}$$

**Theorem 4.2.** *Let $x^* \in \arg\min_x\{F(x)\}$. To solve Problem (3.1) given a starting vector $x_0 \in \mathbb{R}^d$ and any $\eta = \frac{C}{L}$ where $C \in (0, 1]$ is some absolute constant,*

*(a)* $\mathtt{SGD}$ *outputs $\overline{x}$ satisfying $\mathbb{E}[\|\mathcal{G}_{F,\eta}(\overline{x})\|^2] \leq O\big(\frac{L^2\|x_0 - x^*\|^2}{T} + \frac{L\sqrt{\mathcal{V}}\|x_0 - x^*\|}{\sqrt{T}}\big)$.*

*(b) if $T \geq \frac{L}{\sigma}$, then $\mathtt{SGD^{sc}}$ outputs $\overline{x}$ satisfying $\mathbb{E}[\|\mathcal{G}_{F,\eta}(\overline{x})\|^2] \leq O\big(\frac{L\mathcal{V}}{\sigma T}\big) + \big(1 - \frac{\sigma}{L}\big)^{\Omega(T)}\sigma L\|x_0 - x^*\|^2$.*

**Corollary 4.3.** *Hiding $\mathcal{V}, L, \|x_0 - x^*\|$ in the big-O notation, classical SGD finds $x$ with*

$F(x) - F(x^*) \leq O(T^{-1/2})$    $\|\mathcal{G}_{F,\eta}(x)\| \leq O(T^{-1/4})$    *for Problem (3.1), or*

$F(x) - F(x^*) \leq O((\sigma T)^{-1})$    $\|\mathcal{G}_{F,\eta}(x)\| \leq O((\sigma T)^{-1/2})$    *if $f(\cdot)$ is $\sigma$-strongly convex for $\sigma > 0$.*

## 5  An Auxiliary Lemma on Regularization

Consider a regularized objective

$$G(x) := \psi(x) + g(x) := \psi(x) + \big(f(x) + \sum_{s=1}^{S}\frac{\sigma_s}{2}\|x - \widehat{\mathsf{x}}_s\|^2\big) \ , \tag{5.1}$$

---

assumptions. A variant of Theorem 4.1(b) is recorded for the accelerated version of SGD [15], but with a slightly worse rate $T = O\big(\frac{\mathcal{V}}{\sigma T} + \frac{L\|x_0 - x^*\|^2}{T^2}\big)$. If the readers find either statement explicitly stated somewhere, please let us know and we would love to include appropriate citations.

where $\widehat{x}_1, \ldots, \widehat{x}_S$ are fixed vectors in $\mathbb{R}^d$. The following lemma says that, if we find an approximate stationary point $x$ of $G(x)$, then it is also an approximate stationary point of $F(x)$ up to some additive error.

**Lemma 5.1.** *Suppose $\psi(x)$ is proper convex and $f(x)$ is convex and L-smooth. By definition, $g(x)$ is $\widetilde{\sigma}$-strongly convex with $\widetilde{\sigma} := \sum_{s=1}^{S} \sigma_s$. Let $x^*$ be the unique minimizer of $G(y)$ in (5.1), and $x$ be an arbitrary vector in the domain of $\{x \in \mathbb{R}^d : \psi(x) < +\infty\}$. Then, for every $\eta \in \left(0, \frac{1}{L+\widetilde{\sigma}}\right]$, we have*

$$\|\mathcal{G}_{F,\eta}(x)\| \leq \sum_{s=1}^{S} \sigma_s \|x^* - \widehat{x}_s\| + 3\|\mathcal{G}_{G,\eta}(x)\| \ .$$

*Remark* 5.2. Lemma 5.1 should be easy to prove in the special case of $\psi(x) \equiv 0$. Indeed,

$$\|\nabla f(x)\| = \|\nabla g(x) + \sum_s \sigma_s (x - \widehat{x}_s)\| \overset{\text{①}}{\leq} \|\nabla g(x)\| + \sum_s \sigma_s \|x - \widehat{x}_s\|$$

$$\overset{\text{②}}{\leq} \|\nabla g(x)\| + \sum_s \sigma_s \|x^* - \widehat{x}_s\| + \widetilde{\sigma}\|x^* - x\| \overset{\text{③}}{\leq} 2\|\nabla g(x)\| + \sum_s \sigma_s \|x^* - \widehat{x}_s\| \ .$$

Above, inequalities ① and ② both use the triangle inequality; and inequality ③ is due to the $\widetilde{\sigma}$-strong convexity of $g(x)$ (see for instance [23, Sec. 2.1.3]).

*Proof of Lemma 5.1.* See full version. $\qquad\square$

## 6 Approach 3: SGD and Recursive Regularization

In this section, add a logarithmic number of regularizers to the objective, each centered at a different but carefully chosen point. Specifically, given parameters $\sigma_1, \ldots, \sigma_S > 0$, we define functions

$$F^{(0)}(x) := F(x) \quad \text{and} \quad F^{(s)}(x) := F^{(s-1)}(x) + \frac{\sigma_s}{2}\|x - \widehat{x}_s\|^2 \quad \text{for } s = 1, 2, \ldots, S$$

where each $\widehat{x}_s$ (for $s \geq 1$) is an approximate minimizer of $F^{(s-1)}(x)$.

If $f(x)$ is $\sigma$-strongly convex, then we choose $S \approx \log_2 \frac{L}{\sigma}$ and let $\sigma_0 = \sigma$ and $\sigma_s = 2\sigma_{s-1}$. To calculate each $\widehat{x}_s$, we apply SGD$^{\text{sc}}$ for $\frac{T}{S}$ iterations. This totals to a gradient complexity of $T$. We summarize this method as SGD3$^{\text{sc}}$ in Algorithm 3.

If $f(x)$ is not strongly convex, then we regularize it by $G(x) = F(x) + \frac{\sigma}{2}\|x - x_0\|^2$ for some small parameter $\sigma > 0$, and then apply SGD3$^{\text{sc}}$. We summarize this final method as SGD3 in Algorithm 4.

We prove the following main theorem:

**Theorem 3** (SGD3). *Let $x^* \in \arg\min_x\{F(x)\}$. To solve Problem (3.1) given a starting vector $x_0 \in \mathbb{R}^d$ and any $\eta = \frac{C}{L}$ for some absolute constant $C \in (0, 1]$.*

*(a) If $f(x)$ is $\sigma$-strongly convex for $\sigma \in (0, L]$ and $T \geq \frac{L}{\sigma}\log\frac{L}{\sigma}$, then SGD3$^{\text{sc}}(F, x_0, \sigma, L, T)$ outputs $\overline{x}$ satisfying*

$$\mathbb{E}[\|\mathcal{G}_{F,\eta}(\overline{x})\|] \leq O\left(\frac{\sqrt{\mathcal{V}} \cdot \log^{3/2}\frac{L}{\sigma}}{\sqrt{T}}\right) + \left(1 - \frac{\sigma}{L}\right)^{\Omega(T/\log(L/\sigma))} \sigma\|x_0 - x^*\| \ .$$

*(b) If $\sigma \in (0, L]$ and $T \geq \frac{L}{\sigma}\log\frac{L}{\sigma}$, then SGD3$(F, x_0, \sigma, L, T)$ outputs $\overline{x}$ satisfying*

$$\mathbb{E}[\|\mathcal{G}_{F,\eta}(\overline{x})\|] \leq O\left(\sigma\|x_0 - x^*\| + \frac{\sqrt{\mathcal{V}} \cdot \log^{3/2}\frac{L}{\sigma}}{\sqrt{T}}\right) + \left(1 - \frac{\sigma}{L}\right)^{\Omega(T/\log(L/\sigma))} \sigma\|x_0 - x^*\| \ .$$

*If $\sigma$ is appropriately chosen, then we find $\overline{x}$ with $\mathbb{E}[\|\mathcal{G}_{F,\eta}(\overline{x})\|] \leq \varepsilon$ in gradient complexity*

$$T \leq O\left(\frac{\mathcal{V} \cdot \log^3\frac{L\|x_0 - x^*\|}{\varepsilon}}{\varepsilon^2} + \frac{L\|x_0 - x^*\|}{\varepsilon}\log\frac{L\|x_0 - x^*\|}{\varepsilon}\right) \ .$$

*Remark* 6.1. All expected guarantees of the form $\mathbb{E}[\|\mathcal{G}_{F,\eta}(\overline{x})\|^2] \leq \varepsilon^2$ or $\mathbb{E}[\|\mathcal{G}_{F,\eta}(\overline{x})\|] \leq \varepsilon$ throughout this paper can be made into high-confidence bound by repeating the algorithm multiple times, each time estimating the value of $\|\mathcal{G}_{F,\eta}(\overline{x})\|$ using roughly $O(\frac{\mathcal{V}}{\varepsilon^2})$ stochastic gradient computations, and finally outputting the point $\overline{x}$ that leads to the smallest value $\|\mathcal{G}_{F,\eta}(\overline{x})\|$.

---

**Algorithm 3** $\mathtt{SGD3^{sc}}(F, x_0, \sigma, L, T)$

---

**Input:** function $F(x) = \psi(x) + \frac{1}{n}\sum_{i=1}^n f_i(x)$; initial vector $x_0$; parameters $0 < \sigma \leq L$; number
  of iterations $T \geq \Omega\big(\frac{L}{\sigma}\log\frac{L}{\sigma}\big)$. $\quad\quad \diamond\; f(x) = \frac{1}{n}\sum_{i=1}^n f_i(x)$ *is $\sigma$-strongly convex and $L$-smooth*

1: $F^{(0)}(x) := F(x); \widehat{x}_0 \leftarrow x_0; \sigma_0 \leftarrow \sigma;$
2: **for** $s = 1$ **to** $S = \lfloor \log_2 \frac{L}{\sigma} \rfloor$ **do**
3: $\quad \widehat{x}_s \leftarrow \mathtt{SGD^{sc}}\big(F^{(s-1)}, \widehat{x}_{s-1}, \sigma_{s-1}, 3L, \frac{T}{S}\big);$
4: $\quad \sigma_s \leftarrow 2\sigma_{s-1};$
5: $\quad F^{(s)}(x) := F^{(s-1)}(x) + \frac{\sigma_s}{2}\|x - \widehat{x}_s\|^2;$
6: **return** $\overline{x} = \widehat{x}_S.$

---

**Algorithm 4** $\mathtt{SGD3}(F, x_0, \sigma, L, T)$

---

**Input:** function $F(x) = \psi(x) + \frac{1}{n}\sum_{i=1}^n f_i(x)$; initial vector $x_0$; parameters $L \geq \sigma > 0; T \geq 1$.
  $\quad\quad\quad\quad\quad\quad\quad\quad\quad\quad\quad\quad \diamond\; f(x) = \frac{1}{n}\sum_{i=1}^n f_i(x)$ *is convex and $L$-smooth*

1: $G(x) := F(x) + \frac{\sigma}{2}\|x - x_0\|^2;$
2: **return** $\overline{x} \leftarrow \mathtt{SGD3^{sc}}(G, x_0, \sigma, L+\sigma, T).$

---

## 6.1 Proof of Theorem 3

Before proving Theorem 3, we state a few properties regarding the relationships between the objective-optimality of $\widehat{x}_s$ and point distances.

**Claim 6.2.** *Suppose for every $s = 1, \ldots, S$ the vector $\widehat{x}_s$ satisfies*

$$\mathbb{E}\big[F^{(s-1)}(\widehat{x}_s) - F^{(s-1)}(x^*_{s-1})\big] \leq \delta_s \tag{6.1}$$

*where $x^*_{s-1} \in \arg\min_x\{F^{(s-1)}(x)\}$, then,*

*(a) for every $s \geq 1$, $\mathbb{E}[\|\widehat{x}_s - x^*_{s-1}\|]^2 \leq \mathbb{E}[\|\widehat{x}_s - x^*_{s-1}\|^2] \leq \frac{2\delta_s}{\sigma_{s-1}}$,*

*(b) for every $s \geq 1$, $\mathbb{E}[\|\widehat{x}_s - x^*_s\|]^2 \leq \mathbb{E}[\|x^*_s - \widehat{x}_s\|^2] \leq \frac{\delta_s}{\sigma_s}$; and*

*(c) if $\sigma_s = 2\sigma_{s-1}$ for all $s \geq 1$, then $\mathbb{E}\big[\sum_{s=1}^S \sigma_s\|x^*_S - \widehat{x}_s\|\big] \leq 4\sum_{s=1}^S \sqrt{\delta_s\sigma_s}$ .*

*Proof of Claim 6.2.*

(a) $\mathbb{E}[\|\widehat{x}_s - x^*_{s-1}\|]^2 \overset{\textcircled{1}}{\leq} \mathbb{E}[\|\widehat{x}_s - x^*_{s-1}\|^2] \overset{\textcircled{2}}{\leq} \frac{2}{\sigma_{s-1}}\mathbb{E}\big[F^{(s-1)}(\widehat{x}_s) - F^{(s-1)}(x^*_{s-1})\big] \leq \frac{2\delta_s}{\sigma_{s-1}}$. Here, inequality $\textcircled{1}$ is because $\mathbb{E}[X]^2 \leq \mathbb{E}[X^2]$, and inequality $\textcircled{2}$ is due to the strong convexity of $F^{(s-1)}(x)$.

(b) We derive that

$$\sigma_s\|x^*_s - \widehat{x}_s\|^2 \overset{\textcircled{1}}{\leq} \frac{\sigma_s}{2}\|x^*_s - \widehat{x}_s\|^2 + F^{(s)}(\widehat{x}_s) - F^{(s)}(x^*_s) = F^{(s-1)}(\widehat{x}_s) - F^{(s-1)}(x^*_s)$$

$$\overset{\textcircled{2}}{\leq} F^{(s-1)}(\widehat{x}_s) - F^{(s-1)}(x^*_{s-1}) \ .$$

Here, inequality $\textcircled{1}$ is due to the strong convexity of $F^{(s)}(x)$, and inequality $\textcircled{2}$ is because of the minimality of $x^*_{s-1}$. Taking expectation we have $\mathbb{E}[\|x^*_s - \widehat{x}_s\|]^2 \leq \mathbb{E}[\|x^*_s - \widehat{x}_s\|^2] \leq \frac{\delta_s}{\sigma_s}$.

(c) Define $P_t := \sum_{s=1}^t \sigma_s\|x^*_t - \widehat{x}_s\|$ for each $t \geq 0, 1, \ldots, S$. Then by triangle inequality we have

$$P_s - P_{s-1} \leq \sigma_s\|x^*_s - \widehat{x}_s\| + \big(\sum_{t=1}^{s-1}\sigma_t\big)\cdot\|x^*_s - x^*_{s-1}\|$$

Using the parameter choice of $\sigma_s = 2\sigma_{s-1}$, and plugging in Claim 6.2(a) and Claim 6.2(b), we have

$$\mathbb{E}[P_s - P_{s-1}] \leq \sqrt{\delta_s\sigma_s} + \sigma_s\cdot\mathbb{E}\big[\|x^*_s - \widehat{x}_s\| + \|x^*_{s-1} - \widehat{x}_s\|\big] \leq 4\sqrt{\delta_s\sigma_s} \ . \qquad \square$$

*Proof of Theorem 3(a).* We first note that, when writing $f^{(s-1)}(x) = F^{(s-1)}(x) - \psi(x)$, each $f^{(s-1)}$ is at least $\sigma_{s-1}$-strongly convex and $L + \sum_{t=1}^{s-1} \sigma_t \leq 3L$ Lipschitz smooth. Therefore, applying Theorem 4.1(b), we have

$$\mathbb{E}[F^{(s-1)}(\widehat{x}_s) - F^{(s-1)}(x_{s-1}^*)] \leq O\big(\frac{S\mathcal{V}}{\sigma_{s-1}T}\big) + \big(1 - \frac{\sigma_{s-1}}{3L}\big)^{\Omega(T/S)}\mathbb{E}[\sigma_{s-1}\|\widehat{x}_{s-1} - x_{s-1}^*\|^2] \ .$$

If $s = 1$, this means (recalling $\widehat{x}_0 = x_0$ and $x_0^* = x^*$)

$$\mathbb{E}[F^{(0)}(\widehat{x}_s) - F^{(0)}(x^*)] \leq O\big(\frac{S\mathcal{V}}{\sigma_0 T}\big) + \big(1 - \frac{\sigma_0}{L}\big)^{\Omega(T/S)}\sigma_0\|x_0 - x^*\|^2 \ .$$

If $s > 1$, this means

$$\mathbb{E}[F^{(s-1)}(\widehat{x}_s) - F^{(s-1)}(x_{s-1}^*)] \leq O\big(\frac{S\mathcal{V}}{\sigma_{s-1}T}\big) + \big(1 - \frac{\sigma_{s-1}}{L}\big)^{\Omega(T/S)}\mathbb{E}[F^{(s-2)}(\widehat{x}_{s-1}) - F^{(s-2)}(x_{s-2}^*)] \ .$$

Together, this means to satisfy (6.1), it suffices to choose $\delta_s$ so that

$$\delta_s = O\big(\frac{S\mathcal{V}}{\sigma_s T}\big) + \big(1 - \frac{\sigma_0}{L}\big)^{\Omega(sT/S)}\sigma_0\|x_0 - x^*\|^2 \ .$$

Using Lemma 2.3 with $F^{(S-1)}$ and $y = x = \widehat{x}_S$, we have $\frac{\eta}{2}\|\mathcal{G}_{F^{(S-1)},\eta}(\widehat{x}_S)\|^2 \leq F^{(S-1)}(\widehat{x}_S) - F^{(S-1)}(\widehat{x}_S^+) \leq F^{(S-1)}(\widehat{x}_S) - F^{(S-1)}(x_{S-1}^*)$ and therefore

$$\mathbb{E}\big[\|\mathcal{G}_{F^{(S-1)},\eta}(\widehat{x}_S)\|\big]^2 \leq \mathbb{E}\big[\|\mathcal{G}_{F^{(S-1)},\eta}(\widehat{x}_S)\|^2\big] \leq \frac{2\delta_S}{\eta} = O(L\delta_S) \ .$$

Plugging this into Lemma 5.1 (with $G(x) = F^{(S-1)}(x)$) and Claim 6.2(c), we have

$$\mathbb{E}\big[\|\mathcal{G}_{F,\eta}(\widehat{x}_S)\|\big] \leq \mathbb{E}\Big[\sum_{s=1}^{S-1}\sigma_s\|x_{S-1}^* - \widehat{x}_s\| + 3\|\mathcal{G}_{F^{(S-1)},\eta}(\widehat{x}_S)\|\Big] \leq O\Big(\sum_{s=1}^{S-1}\sqrt{\delta_s\sigma_s} + \sqrt{L\delta_S}\Big)$$

$$= O\Big(\sum_{s=1}^{S}\sqrt{\delta_s\sigma_s}\Big) \leq O\Big(\frac{S^{3/2}\mathcal{V}^{1/2}}{T^{1/2}}\Big) + \big(1 - \frac{\sigma_0}{L}\big)^{\Omega(T/S)}\sigma_0\|x_0 - x^*\| \ . \qquad \square$$

*Proof of Theorem 3(b).* Define $G(x) := F(x) + \frac{\sigma}{2}\|x - x_0\|^2$ and let $x_G^*$ be the (unique) minimizer of $G(\cdot)$. Note that $x_G^*$ may be different from $x^*$ which is a minimizer of $F(\cdot)$. Applying Theorem 3(a) on $G(x)$ and Lemma 5.1 with $S = 1$ and $\widehat{x}_1 = x_0$, we have

$$\mathbb{E}[\|\mathcal{G}_{F,\eta}(\overline{x})\|] \leq O\Big(\sigma\|x_0 - x_G^*\| + \frac{\sqrt{\mathcal{V}}\cdot\log^{3/2}\frac{L}{\sigma}}{\sqrt{T}}\Big) + \big(1 - \frac{\sigma}{L}\big)^{\Omega(T/\log(L/\sigma))}\sigma\|x_0 - x_G^*\|$$

Now, by definition $\frac{\sigma}{2}\|x^* - x_0\|^2 - \frac{\sigma}{2}\|x_G^* - x_0\|^2 = (G(x^*) - F(x^*)) + (F(x_G^*) - G(x_G^*)) \geq 0$ so we have $\|x_G^* - x_0\| \leq \|x^* - x_0\|$. This completes the proof. $\qquad \square$

## Acknowledgements

We would like to thank Lin Xiao for suggesting reference [29, Lemma 3.7], an anonymous researcher from the Simons Institute for suggesting reference [25], Yurii Nesterov for helpful discussions, Xinyu Weng for discussing the motivations, Sébastien Bubeck, Yuval Peres, and Lin Xiao for discussing notations, Chi Jin for discussing reference [27], and Dmitriy Drusvyatskiy for discussing the notion of Moreau envelope.

## Footnotes

*The full version of this paper can be found on https://arxiv.org/abs/1801.02982. When this paper was submitted to NeurIPS 2018, the "non-convex SGD" results were not included. We encourage the readers to go to our full version to find out these "non-convex SGD" results.

[2]Nesterov [25] studied $\min_{y \in Q}\{g(y) : Ay = b\}$ with convex $Q$ and strongly convex $g(y)$. The dual problem is $\min_x\{f(x)\}$ where $f(x) := \min_{y \in Q}\{g(y) + \langle x, b - Ay \rangle\}$. Let $y^*(x) \in Q$ be the (unique) minimizer of the internal problem, then $g(y^*(x)) - f(x) = \langle x, \nabla f(x) \rangle \leq \|x\| \cdot \|\nabla f(x)\|$.

[3]Previous authors also refer to this notion as "approximate convex", "almost convex", "hypo-convex", "semi-convex", or "weakly-convex." We call it $\sigma$-nonconvex to stress the point that $\sigma$ can be as large as $L$ (any $L$-smooth function is automatically $L$-nonconvex).

[4]All of the results in this paper apply to the case when $n$ is infinite, because we focus on online methods. However, we still introduce $n$ to simplify notations.

[5]In the special case $\psi(x) \equiv 0$, Theorem 4.1(a) and 4.1(b) are folklore (see for instance [26]). If $\psi(x) \not\equiv 0$, Theorem 4.1(a) is recorded when $\psi(x)$ is Lipschitz or smooth [13], but we would not like to impose such

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
