[Reviews · NeurIPS 2018]

Reviewer 1



Normally, we analyze the convergence rates of gradient descent algorithms for convex problems in terms of f(x)-f(x^*). However, this paper argues that studying the rate of convergence for the gradients is also important. This paper provides a comprehensive and elegant analysis in terms of the gradients for several variants of SGD algorithms, which is the main contributions of this paper. Basically, I think this paper is a wonderful work. The only two weaknesses of this paper I can find are 1. The motivation of this paper is not so strong. 2. It is encouraged to provide the experimental results to support the theoretical results in this paper.

Reviewer 2



The paper addresses stochastic optimisation, with the goal of deriving bounds for finding points with small gradients. They improve previous bounds that were in the order of \epsilon^{-8/3} to \epsilon^{-5/2} and \epsilon^{-2}log^31/ \epsilon. Such work is useful for non=convex optimisation. The authors use ideas by Nesterov to improve convergence rates for accelerated gradient descent (AGD), one being iterations of AGD followed by GD, and the other being AGD after regularizing the objective. The authors apply these ideas to the stochastic setting. The paper is technically sound and correct. It is a little disappointing that the authors do not discuss practical implications of their results. There are no empirical studies, and as such it remains a theory-only paper.

Reviewer 3



This work studies convergence rates of the gradients for convex composite objectives by combining Nesterov’s tricks used for gradient descent with SGD. The authors provide three approaches which differ from each other only slightly and they provide the convergence rates for all the proposed approaches. My comments on this work are as follow: 1. It is indeed important to study convergence rates of gradients especially for non-convex problems. The authors motivate the readers by mentioning this but they assume convexity in their problem set-up. It would have been fine if they were comparing other convergence rates for convex problems as well but I am not sure the existing results they discussed are defined for convex. For instance the work in citation 11 discusses the rates for nonconvex functions. Also, I could not locate the rate \epsilon^{-8/3} in citation 11. I’ll appreciate if the authors clearly mention which result and what assumptions they used from citation 11 in their rebuttal. Same concern applies to the results in citation 15. 2. My general objection about the paper is its representation and organization. I found it really hard to follow the main motivation from a stack of lemmas, corollaries and theorems. I’d suggest authors to narrow their focus to maybe one approach but discuss it in detail. Abstract should also be made more clear. It is not obvious that the manuscript only focuses on convex problems just by reading the abstract. A section of a discussion or a conclusion would also make the manuscript more readable. 3. What is the relationship between \epsilon and \delta between lines 41 and 42? 4. I don’t really understand what “replacing the use of inequality” means in lines 50-51. 5. I am not sure the statement in line 113 about “online” is correct. I understand the term “online” could be used for different things in machine learning but I think it is important to clarify what authors really mean by online setting in this work. Do they mean batch training which requires the entire training data or training with data which appears in a sequential order. Overall, although the work looks interesting in terms of some theoretical convergence results, I think there is still some room to make the manuscript more convincing by working on the representation.